Probable impact of age and hypoxia on proliferation and microRNA expression profile of bone marrow-derived human mesenchymal stem cells

Mohd Ali Norlaily 1
Boo Lily 1
Yeap Swee Keong 2
Ky Huynh 3 4
Satharasinghe Dilan A. 2 5
Liew Woan Charn 2
Ong Han Kiat 1 onghk@utar.edu.my
Cheong Soon Keng 6 7
Kamarul Tunku 8
1 Faculty of Medicine and Health Sciences, Universiti Tunku Abdul Rahman , Cheras, Selangor , Malaysia
2 Institute of Bioscience, Universiti Putra Malaysia , Serdang, Selangor , Malaysia
3 Faculty of Biotechnology and Biomolecular Sciences, Universiti Putra Malaysia , Serdang, Selangor , Malaysia
4 Department of Agriculture Genetics and Breeding, College of Agriculture and Applied Biology, Cantho University , Cantho , Vietnam
5 Faculty of Veterinary Medicine and Animal Science, University of Peradeniya , Peradeniya , Sri Lanka
6 Faculty of Medicine and Health Sciences, Universiti Tunku Abdul Rahman , Kajang, Selangor , Malaysia
7 Cryocord Sdn Bhd , Cyberjaya , Malaysia
8 Tissue Engineering Group (TEG), National Orthopaedic Center of Excellence for Research and Learning (NOCERAL), Faculty of Medicine, Universiti Malaya , Kuala Lumpur , Malaysia
Goswami Chandan
Electronic publication date: 2016 Jan 14
Publication date: 2016
Volume: 4
Electronic Location ID: e1536
Received 2015 Sep 1; Accepted 2015 Dec 5
Copyright: © 2016 Mohd Ali et al.
Copyright year: 2016
Copyright holder: Mohd Ali et al.
License: This is an open access article distributed under the terms of the Creative Commons Attribution License, which permits unrestricted use, distribution, reproduction and adaptation in any medium and for any purpose provided that it is properly attributed. For attribution, the original author(s), title, publication source (PeerJ) and either DOI or URL of the article must be cited.
License URL: https://creativecommons.org/licenses/by/4.0/

Keywords: Hypoxia, Differentiation, Age, Proliferation, Bone-marrow, Mesenchymal stem cells, Next generation sequencing, MicroRNA, Differentiation

Funding: University of Malaya’s HIR-MoE Reference number–UM.C/625/1/HIR/MOHE/CHAN/03, account number–A000003-50001 This research was supported by University of Malaya’s HIR-MoE Grant (Reference number–UM.C/625/1/HIR/MOHE/CHAN/03, account number–A000003-50001). The funders had no role in study design, data collection and analysis, decision to publish, or preparation of the manuscript.

==============================
Decline in the therapeutic potential of bone marrow-derived mesenchymal stem cells (MSC) is often seen with older donors as compared to young. Although hypoxia is known as an approach to improve the therapeutic potential of MSC in term of cell proliferation and differentiation capacity, its effects on MSC from aged donors have not been well studied. To evaluate the influence of hypoxia on different age groups, MSC from young (<30 years) and aged (>60 years) donors were expanded under hypoxic (5% O2) and normal (20% O2) culture conditions. MSC from old donors exhibited a reduction in proliferation rate and differentiation potential together with the accumulation of senescence features compared to that of young donors. However, MSC cultured under hypoxic condition showed enhanced self-renewing and proliferation capacity in both age groups as compared to normal condition. Bioinformatic analysis of the gene ontology (GO) and KEGG pathway under hypoxic culture condition identified hypoxia-inducible miRNAs that were found to target transcriptional activity leading to enhanced cell proliferation, migration as well as decrease in growth arrest and apoptosis through the activation of multiple signaling pathways. Overall, differentially expressed miRNA provided additional information to describe the biological changes of young and aged MSCs expansion under hypoxic culture condition at the molecular level. Based on our findings, the therapeutic potential hierarchy of MSC according to donor’s age group and culture conditions can be categorized in the following order: young (hypoxia) > young (normoxia) > old aged (hypoxia) > old aged (normoxia).

Introduction

Mesenchymal stem cells derived from bone marrow (BM-MSCs) are adult multipotent stem cells with the self-renewing capacity and the ability to differentiate into cells of various connective tissue lineages. They are regarded as a promising and potential alternative source in the repair of many cells and tissues due to its multilineage differentiation capability into not only mesoderm but also ectodermic and endodermic lineages such as osteoblasts, chondrocytes, adipocytes, neurocytes and myoblasts (Wei et al., 2013; Zhou et al., 2008). MSCs are clinically used in engraftment of post-transplantation or as gene therapy vehicles in osteogenesis imperfecta because of their immunosuppressive capacity and are widely used in the treatment of cardiac disorder, musculoskeletal and cancer (Baxter et al., 2004; Mognetti et al., 2013; Mohyeldin, Garzon-Muvdi & Quinones-Hinojosa, 2010; Oliveira et al., 2012). Outstanding features of MSC are that it can be easily obtained from various sources of adult tissue (adipose and bone-marrow) and postnatal tissues (Wharton-jelly and umbilical cord) and can be expanded in vitro (Oliveira et al., 2012; Samsonraj et al., 2013).

The challenge for MSC-therapy is that it requires high yield and good quality of stem cells. In order to get sufficient yield, stem cells need to be expanded under prolonged passage, which can lead to deterioration of its self-renewal and differentiation capacity. Prolonged passage has been reported to be directly linked with the shortening of telomere length that leads to decrease in cell proliferation and increase of senescence (Samsonraj et al., 2013). Age is another factor that is associated with progressive loss of cell proliferation resulting in cellular senescence. Biological markers of cellular senescence were highly expressed in MSC from aged donors along with oxidative damage indicators (Reactive oxygen species (ROS) and nitric oxide) (Stenderup et al., 2003; Stolzing et al., 2008). The acceleration of senescence was related with the decrease of cell proliferation and their life span as well as contributing to the accumulation of DNA damage leading to stem cell exhaustion (Rube et al., 2011). These marked the compromised quality of MSCs with age as well as prolonged passage under normal condition. Despite that, in the case of autologous MSC transplantation for severe autoimmune diseases and old age related diseases in aged patients, the use of MSC of aged donor is still highly in demand (Wang et al., 2013; Wei et al., 2013).

Alternatively, low oxygen content (5% O2), referred as hypoxia, is regarded as a characteristic feature of their microenvironment in vivo that provides microenvironmental stresses to stem cells, allowing them to escape senescence pathways resulting in an increased proliferation rate (Efimenko et al., 2011). Hypoxia is known to stimulate pro-angiogenic effects in stem cells while maintaining the telomere length (Efimenko et al., 2011; Oliveira et al., 2012). Most of our knowledge concerning MSC’s differentiation and proliferation potential reduction by age had been obtained from cell cultured under normal oxygen as well as hypoxia conditions. Although the hypoxic condition has been demonstrated as a mode to improve the therapeutic potential of MSC (Cicione et al., 2013; Guan et al., 2012; Peterson et al., 2011; Rosova et al., 2008), little is known about its effect on a specific age group, especially in the involvement and the interplay of miRNAs in the proliferation and renewal properties. As an important class of regulatory noncoding RNAs, miRNAs have been shown to play important roles in the committed differentiation and self-renewal of embryonic stem cells and adult stem cells (Huang, Le & Giaccia, 2010). Interestingly, the miRNA expression patterns in MSC when subjected to various factors and culture conditions such as hypoxia and serum deprivation differ considerably (Nie et al., 2011). By preventing the translation of mRNAs, miRNAs are predicted to regulate almost all cell’s biological processes, ranging from cell renewal, differentiation, migration, tumorigenesis to cell death, aging and senescence (Di Leva & Croce 2013; Nie et al., 2011; Tome et al., 2011).

In this work, we investigated the influence of donor’s age on the miRNA expression profiles of BM-MSCs expanded under hypoxic culture and discovered several miRNAs that may potentially act as key regulators.

Materials and Methods

Hypoxic culture of human BM-derived MSCs

The primary BM-MSC collection was obtained from Cryo Cord Sdn Bhd and the research approved by the UTAR Scientific and Ethical Review Committee (U/SERC/14/2012), in compliance with international guidelines regarding the use of primary bone marrow stem cell for research and signed consent were obtained prior to study. BM-derived MSC was prepared as described previously (Efimenko et al., 2011). Six subjects (F/M = 4/2/; Age = 19–80 years old) are with no metabolic disease, inherited diseases and other diseases which may affect the current study. MSCs from young (≤30 years old) and aged (≥60 years old) donors were (n = 3/group) maintained in Dulbecco’s modified Eagle’s medium (DMEM, Sigma, USA) supplemented with 10% fetal bovine serum (GIBCO, USA) at 37 °C, in 5% CO2 and 20% O2. Hypoxic (Hx) condition was attained by culturing cells in a gas mixture composed of 94% N2, 5% CO2, and 5% O2 (Ueyama et al., 2012). Cumulative population doubling (CPD) was counted using Typan blue assay. The population doubling was determined using the formula x = log2 (N2)/(N1)/log2, where N2 is the number of harvested cells, and N1 is the number of seeded cells. To calculate the CPD, population doubling in each passage was determined and compared with the population doubling of earlier passages (Stolzing et al., 2008).

MSC characterizations

Immunophenotyping of MSCs

Identification of MSCs was performed using CD90, CD44, CD105 and CD19 MSC surface markers from Flow Cellect kit (Merck, Kenilworth, NJ, USA). Cells were incubated with 18 μL of antibody working cocktail solution (Anti-human CD105-conjugated with FITC, anti-human CD90-conjugated with PE, anti-human CD44-conjugated with Alexa fluor 647, anti-human CD19-conjugated with PECy5) for 30 minutes in the dark at 4 °C and acquired using a BD FACS Canto TM II Flow Cytometer (Becton Dickinson, Franklin Lakes, NJ, USA).

Multipotent differentiation

The multipotency of MSCs were evaluated using adipogenic and osteogenic assays (Pandey et al., 2011). Both MSC groups expanded under normoxic and hypoxic conditions were cultured in low-glucose DMEM containing 10% fetal bovine serum, antibiotics, 100 U/ml penicillin and 100 μg/ml streptomycin and 2 mM L-glutamine until 90% confluent. For in vitro differentiation into osteoblasts, cells were induced with osteogenic induction medium (OIM), composed of DMEM supplemented with 10% FBS, 50 μg/mL ascorbate-2 phosphate, 108M dexamethasone, and 10 mM β-glycerophosphate. For in vitro differentiation into adipocytes, cells were induced with adipogenic induction medium (AIM), composed of DMEM supplemented with 10% FBS, 50 μg/mL ascorbate-2 phosphate, 107 M dexamethasone, 50 μM indomethacin, 0.45 mM 3-isobutyl-1-methyl-xanthine (IBMX) and 10 μg/mL insulin. Change of medium was performed every 3 days until 21 days when the matrix mineralization and lipid droplets were fully formed.

β-galactosidase staining

Confluent cells were fixed with 2% formaldehyde for 5 minutes and incubated with senescence-associated β-Gal (SA-β-Gal) chromogenic substrate solution (Cell Signaling Technology, Beverly, MA, USA). The mean percentage of cells expressing β-galactosidase was quantified using ImageJ with an average of 200 cells.

Next Generation Sequencing (NGS) of miRNA

Total RNAs including small RNAs were isolated from cells at passage 15 using a miRNeasy kit (Qiagen, Valencia, CA, USA). The next generation sequencing (NGS) library was prepared using TruSeq small RNA sample prep kit (Illumina, San Diego, CA, USA). Briefly, 3′ and 5′ RNAs adapters were ligated to small RNAs molecules, reverse-transcribed and PCR amplified. Prior to amplification, indices for sample multiplexing were incorporated. Samples in the same multiplex group were pooled and run on a 6% SDS-PAGE and validated using an Agilent Technologies Bioanalyzer high sensitivity DNA chip. Data were analyzed using CLC Genomic Workbench 7.0 and sequence with length less than 17 or more than 27 were removed. miRNA sequences were aligned to human genome (Homo sapiens GRCh 37.57) and known miRNA database (miRBase-19) of both mature and miRNA precursors (Noren Hooten et al., 2013).

Pathway analysis

Target genes were identified using a target prediction program miRDB. The identified genes were further analyzed using the web-based DAVID Bioinformatic Resource functional annotation tool (http://david.abcc.ncifcrf.gov/) for pathway analysis to identify the gene ontology (GO) and significantly enriched pathways using the Kyoto Encyclopedia of Genes and Genomes (KEGG).

MicroRNAs quantification by real-time PCR analysis

Five miRNAs were selected to validate NGS data. cDNA synthesis and real-time qPCR were performed with miRCURY LNA Universal RT microRNA PCR (Exiqon, Denmark). In brief, RNAs were tailed with a poly (A) sequence at their 3′end and reverse-transcribed into cDNA using a universal poly (T) primer with a 3′end degenerate anchor and a 5′end universal tag. The cDNA products were quantified using SYBR green based real time PCR and locked nucleic acid (LNA) enhanced miRNA specific primers. The qPCR analysis was run on a CFX96™ touch thermocycler (Biorad) (Jensen et al., 2011). Normalization was done using the average value of miR-200a, miR-122-5p and miR-16 evaluated using geNorm algorithms. Primer sequences for qPCR are listed in Table 1.

Table 1 The accession number and target sequence of the primers used in the quantitative real-time PCR assay.

Name	Accession number	Target sequence	
hsa-miR-200a-3p (Reference)	MIMAT0000682	UAACACUGUCUGGUAACGAUGU	
hsa-miR-122-5p (Reference)	MIMAT0000421	UGGAGUGUGACAAUGGUGUUUG	
hsa-miR-34b-3p	MIMAT0004676	CAAUCACUAACUCCACUGCCAU	
hsa-miR-210	MIMAT0000267	CUGUGCGUGUGACAGCGGCUGA	
hsa-miR-19b-3p	MIMAT0000074	UGUGCAAAUCCAUGCAAAACUGA	
hsa-miR-33a-5p	MIMAT0000091	GUGCAUUGUAGUUGCAUUGCA	
hsa-miR-21-5p	MIMAT0000076	UAGCUUAUCAGACUGAUGUUGA	

Statistical analysis

Data are expressed as means ± standard deviations. Statistical analyses were performed using one-way analysis of variance (ANOVA), FDR correction and Duncan’s post hoc test. The results were taken to be significant at a probability level of p-values <0.05.

Results

Comparison of BM-MSC of young and aged donors under hypoxic and prolonged passage conditions

MSC surface marker expressions showed that CD105, CD90 and CD44 were strongly expressed in young MSC whereas in aged MSC, the markers were present but the expression level was slightly lower. Meanwhile, the expression of CD19 surface marker was not detectable on either young or aged MSC (Fig. 1A) confirming that both groups exhibited phenotype common to MSC. In normal oxygen condition, the proliferation rate of MSC remained relatively high during low passage. As the number of passage increased, the proliferation rate of aged MSC declined significantly. Meanwhile, MSCs subjected to hypoxic treatment exhibited relatively higher proliferation rate in both groups. Overall, aged MSC had significantly lower cumulative population doubling (CPD) compared to young MSC (Fig. 1B), which further showed a comparatively reduced proliferation rate, resulting in growth arrest or early senescence. This was further supported based on morphological assessment where young MSC displayed fibroblast-like morphology with a long-spindle shape whereas substantial alteration of morphology such as loss of their characteristic spindle-shaped morphology and increase in cytoplasmic volume and size (spread-out and polygonal shape) in aged MSC (Fig. 1C). The positive staining of Alizarin Red and Oil Red O confirmed the multilineage differentiation of hypoxic-treated MSCs into osteocytes and adipocytes (Fig. 1D). A decline in the formation of calcium deposition and the number of cells with lipid droplets in hypoxic conditions of MSC from aged donors compared to the young donors was noticed indicating the reduction in their differentiation ability. Differentiation ability was further confirmed with differentiation markers (aP2, adiponectin, RUNX2 and osteopontin) using RT-PCR (Fig. 1E). Subsequently, percentage of MSC positive for SA-β-gal increased tremendously with age as a substantial expression of blue color SA-β-gal positive cells were detected in aged MSC compared to young (Fig. 1F). Furthermore, cells expanded under hypoxic condition had a lower senescence expression relative to the cells expanded under normal condition.

Figure 1 MSC characterization.

(A) Immunophenotyping of BM-MSC from young and aged donors (n = 3). Representative graphs were all positive for CD105, CD90, CD44 and negative for CD19; (B) Cumulative population doubling (CPD) of MSC expanded under normal and hypoxic conditions at p15; (C) BM-MSC morphology during culture expansion under normal and hypoxic at low passage (P5) and high passage (P15). Arrow indicates differentiated cells that have changed in size and morphology with evidence of cytoplasm spreading (n = 3). (Magnification of × 10 scale bar = 100 μm); (D) Mesodermal differentiation of young and aged MSCs post exposure to hypoxia. Cells from young and aged MSCs (p15) were cultured in standard medium (control), osteogenic induction medium and adipogenic induction medium respectively. (Magnification of = 20 scale bar = 100 μm); (E) Representative reverse-transcription polymerase chain reaction run on 2% of agarose gel showing expression of aP2 and adiponectin by adipocytes and RUNX2 and osteopontin by osteocytes cells, induced from human bone-marrow mesenchymal stem cells. Control cells without induction medium treatment either did not show any expression or showed lower level of expression of the above markers than the differentiated cells; (F) Senescence-associated βGal activity in MSC of normal vs Hypoxic at p15. Results are considered as significantly changed when *p < 0.05 using one-way ANOVA.

miRNAs expression profiles in hypoxia-treated MSC

To screen miRNAs that might be involved in hypoxia implicated cell fate commitment leading to increase of cellular proliferation and differentiation potential, next gene sequencing using Illumina platform was used to compare miRNA expression profiles of MSC between young and aged donors (Fig. 2). On average, 3.2 million effective reads were obtained from young MSC and 4.0 million effective reads from aged MSC with a quality score (>Q30) of 93.3%. The average length of the detected sequences was 21.6 nucleotides after the removal of the 5′ and 3′ adapter sequences. An average of 46.8% and 38.6% small RNAs were obtained from young and aged MSC respectively using miRBase-19 database. Meanwhile, 53.2% and 61.4% small RNAs were obtained from young and aged MSC respectively using Homo sapiens GRCh 37.57 library. Approximately, 27–34% of the small RNAs matched with the known human miRNAs, and the remaining unmatched sequences (65–72%) were classified as novel miRNAs. All the data can be found under GEO accession number GSE67630. The distribution of miRNAs that are commonly upregulated and downregulated in young and old MSC donors were illustrated in Venn-diagram (Figs. 3A and 3B). The overlap miRNAs from each group might contribute to similar phenotype observed in the study. Relative distances between the miRNA profiles of different age groups and hypoxic treatment populations were shown in principal component analyses (PCA) plot (Fig. 3C). A total of nine and two miRNAs with fold change >2 were up-regulated in young and aged MSC respectively. Meanwhile, four and 31 miRNAs with fold change >2 were down-regulated in young and aged MSC respectively (Table 2).

Figure 2 NGS Workflow.

Workflow of a miRNA next generation sequencing procedure and data mining.

Figure 3 Venn-diagram and PCA plot.

(A) Venn-diagram illustrating groups of miRs that are commonly upregulated and downregulated in young donors, (B) Venn-diagram illustrating groups of miRs that are commonly upregulated and downregulated in old donors, (C) Principal component analyses indicating the relative distances between the miRNA profiles of different age groups and hypoxic treatment populations.

Table 2 List of differentially expressed miRNAs of MSC from young and aged donors cultured under hypoxic (Hx) culture condition relative to normal (nx) culture condition.

miRNA of young MSC	Fold change >2 (Hx/nx)	p-value	
mir-210	13.64	0.0119	
mir-423	2.58	0.0105	
mir-1468	2.30	0.0243	
mir-21	2.26	0.0197	
mir-3605	2.21	0.0061	
mir-625	2.20	0.0709	
mir-155	2.08	0.0071	
mir-3065	2.07	0.0082	
mir-138-1	2.06	0.0034	
mir-424	−2.85	0.0073	
let-7i	−3.22	0.0038	
mir-655	−3.67	0.0323	
mir-33a	−4.15	0.0182	
miRNA of aged MSC	Fold change >2 (Hx/nx)	p-value	
mir-7977	2.55	0.0350	
mir-195	2.15	0.0061	
mir-19b	−2.58	0.0280	
mir-21	−2.61	0.0089	
mir-99a	−2.74	0.0168	
mir-708	−3.17	0.0030	
mir-1185-1	−3.23	0.0048	
mir-590	−3.24	0.0102	
mir-455	−3.46	0.0039	
mir-374a	−3.53	0.0315	
mir-381	−3.53	0.0340	
mir-126	−3.87	0.0019	
mir-196a-2	−3.99	0.0340	
mir-136	−4.03	0.0081	
mir-100	−4.29	0.0062	
mir-34b	−4.33	0.0159	
mir-542	−4.35	0.0277	
mir-376a-1	−4.35	0.0307	
mir-1197	−4.53	0.0053	
let-7f-2	−4.69	0.0045	
mir-1228	−4.70	0.0091	
mir-2355	−4.73	0.0143	
mir-148b	−4.79	0.0729	
mir-33a	−4.84	0.0295	
mir-551b	−4.88	0.0013	
mir-483	−6.29	0.0061	
mir-29a	−6.40	0.0003	
mir-378a	−6.56	0.0114	
mir-376b	−8.12	0.0039	
mir-561	−8.28	0.0006	
mir-29b-1//mir-29b-2	−9.30	0.0340	
mir-193a	−11.88	0.0315	
mir-627	−15.00	0.0483	
Note:

(P < 0.05) with at least two-fold change.

Target genes, gene ontology (GO) and KEGG pathways of differentially expressed miRNAs

To assess the possible biological impact of the differentially expressed miRNAs in MSC, the predicted target genes of miRNAs (fold change >2, p > 0.05) were analyzed. A total of 2,118 and 3,617 potential targets were identified through this process for differentially expressed miRNAs in young and aged MSCs respectively. In young MSC, the putative target genes of known miRNAs appeared to be involved in a broad range of GO analysis (biological processes) with most of them related to transcriptional categories such as regulation of transcription and inflammatory response to stress (GO:0043619, GO:0043618) (Table 3A), whereas in aged MSC, negative regulation of epithelial to mesenchymal transition (GO:0010719) and differentiation (GO:0010771) were among the top 10 most prominent terms (Table 3B).

Table 3 The top 10 of highly enriched gene ontology (GO) terms of biological processes in predicted targets miRNAs (fold change >2) of MSC from young and aged donors cultured under hypoxic (Hx) culture condition relative to normal (nx) culture condition.

Function	p-value	Count	%	Fold enrichment	
(A) Top 10 of highly enriched GO (biological processes) in MSC from young donors	
Regulation of transcription from RNA polymerase II promoter in response to oxidative stress GO:0043619	3.70E–02	3	0.2	9	
Polyol transport GO:0015791	9.50E–03	4	0.3	8	
Synaptic vesicle maturation GO:0016188	5.80E–02	3	0.2	7.2	
Protein palmitoylation GO:0018345	5.80E–02	3	0.2	7.2	
Healing during inflammatory response GO:0002246	5.80E–02	3	0.2	7.2	
Regulation of transcription from RNA polymerase II promoter in response to stress GO:0043618	5.80E–02	3	0.2	7.2	
Nuclear pore organization GO:0006999	5.80E–02	3	0.2	7.2	
Regulation of transcription in response to stress GO:0043620	5.80E–02	3	0.2	7.2	
Definitive hemopoiesis GO:0060216	1.60E–02	4	0.3	6.9	
Embryonic camera-type eye morphogenesis GO:0048596	2.30E–02	4	0.3	6	
(B) Top 10 of highly enriched GO (biological processes) in MSC from aged donors	
Negative regulation of epithelial to mesenchymal transition GO:0010719	4.00E–02	2	0.6	49.2	
Voluntary musculoskeletal movement GO:0050882	4.00E–02	2	0.6	49.2	
Negative regulation of cell morphogenesis involved in differentiation GO:0010771	6.00E–02	2	0.6	32.8	
Epithelial cell maturation GO:0002070	9.70E–02	2	0.6	19.7	
Regulation of calcium ion-dependent exocytosis GO:0017158	1.30E–02	3	0.8	16.4	
Vesicle transport along microtubule GO:0047496	1.30E–02	3	0.8	16.4	
Regulation of epithelial to mesenchymal transition GO:0010717	1.30E–02	3	0.8	16.4	
Protein ubiquitination during ubiquitin-dependent protein catabolic process GO:0042787	4.50E–03	4	1.1	11.6	
Cerebral cortex cell migration GO:0021795	3.20E–02	3	0.8	10.5	
Bone resorption GO:0045453	3.60E–02	3	0.8	9.8	

The signaling pathways that were found to be involved in KEGG analysis of MSC from young donors included ErbB signaling, melanogenesis, Wnt, MAPK and GnRH signaling pathways (Table 4). Meanwhile, cell cycle, Wnt, p53 and calcium signaling pathway were the most enriched pathways seen in MSC from aged donors. Cell cycle and p53-signaling pathways are involved in replicative senescence leading to permanent growth arrest. Moreover, pathways involved in various signal transduction and cell-cell interactions such as Wnt and MAPK signaling pathways and others were significantly enriched in both young and aged MSC.

Table 4 KEGG pathways enrichment of miRNA target predicted genes (fold change >2) of MSC from young and aged donors cultured under hypoxic (Hx) culture condition relative to normal (nx) culture condition.

Pathways	Count	%	P-value	Fold enrichment	Benjamini	FDR	
(A) KEGG pathways enrichment of miRNA target predicted genes of young MSC	
GnRH signaling pathway	17	1.1	1.30E-03	2.4	2.00E-01	1.50E + 00	
Melanogenesis	16	1	3.80E-03	2.3	2.00E-01	4.50E + 00	
Vascular smooth muscle contraction	16	1	1.20E-02	2	4.00E-01	1.40E + 01	
Type II diabetes mellitus	9	0.6	1.60E-02	2.7	4.30E-01	1.80E + 01	
Gap junction	13	0.9	2.20E-02	2.1	4.20E-01	2.40E + 01	
MAPK signaling pathway	29	1.9	2.20E-02	1.5	4.70E-01	2.30E + 01	
Wnt signaling pathway	18	1.2	3.70E-02	1.7	5.20E-01	3.70E + 01	
ErbB signaling pathway	12	0.8	4.20E-02	1.9	5.20E-01	4.10E + 01	
Adherens junction	11	0.7	4.40E-02	2	5.00E-01	4.20E + 01	
Long-term potentiation	10	0.7	4.90E-02	2.1	5.10E-01	4.60E + 01	
(B) KEGG pathways enrichment of miRNA target predicted genes of aged MSC.	
Insulin signaling pathway	10	2.8	1.70E-03	3.6	1.90E-01	2.00E + 00	
Wnt signaling pathway	9	2.5	1.20E-02	2.9	3.10E-01	1.30E + 01	
p53 signaling pathway	6	1.7	1.30E-02	4.2	2.60E-01	1.40E + 01	
B cell receptor signaling pathway	6	1.7	1.90E-02	3.8	3.10E-01	2.00E + 01	
Cell cycle	7	2	4.40E-02	2.7	3.20E-01	4.00E + 01	
MAPK signaling pathway	11	3.1	4.80E-02	2	3.20E-01	4.30E + 01	
Calcium signaling pathway	8	2.2	7.10E-02	2.2	3.80E-01	5.70E + 01	
Type II diabetes mellitus	4	1.1	7.20E-02	4.1	3.60E-01	5.80E + 01	
mTOR signaling pathway	4	1.1	9.10E-02	3.7	4.20E-01	6.70E + 01	

Quantitative RT-PCR validation of selected microRNA

To validate miRNA expression changes in MSC after hypoxic treatment, we performed the RT-qPCR of the five highly dysregulated miRNAs (miR-21, miR-34b, miR-210, miR-19b, miR-33a). The miRNAs were selected for validation based on statistical significance (P < 0.05) and their key role in biological or signaling pathways altered in hypoxic treatment with respect to control (normal) culture condition. As illustrated in Fig. 4 and Supplementary Table 4, the NGS data correlated well with the RT-qPCR results, indicating the reliability of sequencing based expression analysis.

Figure 4 qPCR validation and correlation plot.

Reproducibility of NGS and comparison of miRNA expression between NGS and qPCR analysis. (A) qPCR analysis in MSC of young and old age donors, (B) Correlation plot comparing the Fold change values of qPCR with the log2 of the NGS in young donors, (C) Correlation plot comparing the Fold change values of qPCR with the log2 of the NGS in old donors. Corresponding R2 values were determined by linear regression analysis.

Discussion and Conclusions

The latest profiling platform, NGS was used to generate miRNA expression profiles and further elaborate the relationship of hypoxia-inducible miRNA in young and aged MSC. The introduction of high throughput sequencing approaches has provided opportunities to generate inclusive sequencing data for the identification and quantification of reproducible known and novel miRNAs compared to previous approaches such as qPCR and microarray. In normal culture condition of prolonged passage, aged MSC displayed a great reduction in viability causing the cells to undergo morphological transformation, senescence and detachment leading to cell death. This is in contrast with young MSC, which retained their MSC characteristics. This finding is showing that MSC isolated from old donors tend to exhibit slower progress on their proliferation, expansion and differential capacity relative to young MSC, similar to a previous study reported by other researchers (Choudhery et al., 2014). Meanwhile, MSC cultured under hypoxic condition showed enhanced self-renewal and proliferation capacity in both age groups compared to normal condition. Nevertheless, young MSC performed better than the aged MSC.

Research on the roles of hypoxia-related miRNAs in age-related MSC will have great significance on therapeutic applications of MSC. We therefore hypothesize that these 2 groups of MSCs (young and old age) have distinct etiologies and could present potentially different pathways that might involve and converge in a common biological processes particularly in the proliferation and differentiation processes. Sequencing results revealed differential expression of several hypoxia-inducible miRNAs in young MSC (Table 2). The up-regulation of miR-210, miR-21 and miR-155 as well as down-regulation of let-7i and miR-33a, were among those that were predicted to target mRNAs associated with transcriptional activity leading to enhanced cell proliferation, survival and migration as well as a decrease in growth arrest and apoptosis (Bruning et al., 2011; Clark et al., 2014). Meanwhile, miR-627 (negative regulator of proliferation) and miR-193a (anti-apoptotic regulator) were significantly down-regulated in aged MSC. Moreover, miR-193a was previously found to inhibit ErbB protein translation, which subsequently inhibited cell proliferation and promoted apoptosis in tumor. ErbB signal is important for stem cells development and maintenance (Liang et al., 2015). Our finding demonstrated the involvement of hypoxia in inhibiting miR-627 and miR-193a expression in aged MSC and consequently contributed to increase in proliferation activity. As a whole, this study indicates the possible link of hypoxia-inducible miRNAs in regulating survival and self-renewal, thereby improving the proliferation activity of young and aged MSCs relative to normal oxygen culture condition.

Furthermore, it was shown that the age of donor’s MSC played a role in the capacity to differentiate into certain lineages. The reduction of adipocyte and osteocyte differentiation was clearly seen in prolonged passage of aged MSC. Previous research reported that shortening of telomere length might be the cause of reduction in differentiation and self-renewal capacity (Zhou et al., 2008). The miRNAs identified in this study that were up-regulated by hypoxia (miR-210, miR-29b) in young MSC and down-regulated (miR-196a, miR-148b) in aged MSC were shown to be involved in promoting osteocytes lineage differentiation. Similarly, miRNAs that were involved in adipocyte differentiation (miR-21) was found to be up-regulated in young and down-regulated in aged MSC. A set of highly down-regulated miRNAs in aged MSC, notably miR-29a and miR-29b (regulated chondrogenic differentiation, cartilage formation and neuronal differentiation) and miR-561 (post-transcriptional regulation of mRNAs) were reported to be associated with human embryonic and other stem cells (Duan et al., 2014; Guerit et al., 2014). Induction and inhibition of these hypoxia-inducible miRNAs may explain their role in the differentiation capacity of MSC. Based on the previous findings and our results, it is highly probable that hypoxic conditioning may promote MSC survival via regulation of various miRNAs.

Bioinformatic analysis and target prediction have been used as the main methods to explore the function of miRNAs. The genes possibly regulated by hypoxia-inducible miRNAs are involved in both tumorigenesis and stem cell maintenance such as cell survival, differentiation, angiogenesis, apoptosis, cell cycle regulation, mitochondrial metabolism as well as DNA damage repair (Huang, Le & Giaccia, 2010; Kondo et al., 2001; Tsai et al., 2011). MiR-210 and miR-21 were reported to regulate cell proliferation by targeting fibroblast growth factor receptor-like 1 (FGFRL1) (Table 5) (Huang, Le & Giaccia, 2010; Tsuchiya et al., 2011). MiR-155 was found to be up-regulated in young MSC and inhibition of miR-155 in senescence cells was reported to cause an elevated level of a protein involved in the TP53 growth arrest pathway (TP53INP1) (Table 4B) (Faraonio et al., 2012). Young MSC cultured under hypoxia which showed an up-regulation of miR-155 was able to maintain its proliferative activity and has yet to reach the growth arrest and senescence stage. The results indicated that hypoxia-inducible miRNAs are predicted to target mRNAs and transcriptional activity leading to enhanced cell proliferation and migration as well as a decrease in growth arrest and apoptosis which may serve as a clinical marker for the proliferation and expansion in MSC.

Table 5 The top 5 of highly predicted target genes of differentially expressed miRNAs in young and aged MSCs identified using target prediction program miRDB.

Gene symbol (Young MSC)	Gene description	
IGF2	Insulin-like growth factor 2 (somatomedin A)	
FGFRL1	Fibroblast growth factor receptor-like 1	
PTPN21	Protein tyrosine phosphatase, non-receptor type 21	
ISCU	Iron-sulfur cluster scaffold homolog (E. coli)	
SLC25A26	Solute carrier family 25, member 26	
Gene symbol (Aged MSC)	Gene description	
FGF2	Fibroblast growth factor 2 (basic)	
MTMR3	Myotubularin related protein 3	
CASK	Calcium/calmodulin-dependent serine protein kinase (MAGUK family)	
WEE1	WEE1 homolog (S. pombe)	
CCNE1	Cyclin E1	

A previous finding reported that adipogenic and osteogenic differentiation capacity of MSC diminishes with age (Yu et al., 2011). MiR-210 and miR-29a have been identified as positive regulators of osteocyte differentiation by inhibiting the TGF-beta/activin signaling pathway through the inhibition of the target gene AcvR1b in MSC (He et al., 2013). In our studies, a similar trend could be observed where miR-210 was predicted to target the putative gene TGF-beta that might be involved in promoting osteogenesis and adipogenesis in young MSC. Meanwhile miR-29a was highly down-regulated in aged MSC in which the differentiation into osteocytes was shown to be inhibited.

Overall, significantly differentially expressed miRNAs in both young and aged MSCs after hypoxic exposure were mostly shown to target putative genes (Table 5) involving multilineage differentiation, proliferation as well as apoptosis (Table 3). Each differentially expressed miRNAs (up-regulated and down-regulated) with hundreds of predicted target putative genes that were linked with multiple GO and KEGG pathways were combined and hypoxia-related miRNAs were shown and presumed to be the main drivers and regulators of proliferation and differentiation favoring young MSC.

Here we reported the different profiles of hypoxia-inducible miRNA signatures between young and aged MSCs, thereby providing additional information on possible link of miRNAs associated with hypoxic condition that increased the proliferation of both young and aged MSCs. These miRNAs can be further used as molecular markers for screening the quality of MSC. In conclusion, our results showed that both young and aged MSCs cultured under hypoxia performed better than normal condition. In this respect, hypoxia enhanced the self-renewal and multi-potent differentiation potential of BM-MSC especially in aged (compromised) MSC. This study not only provided further understanding on the likely effects of age in inducing biological changes in MSC, but more importantly the molecular changes critical to the successful application of MSC in regenerative medicine.

Supplemental Information

Supplemental Information 1 Supplementary data for figure 1A, 1B, 1E and 3.xlsx.

Tables of raw data for Figure 1: MSCs characterization; (A) Immunophenotyping of BM-MSC from young and aged donors (n = 3). Representative graphs were all positive for CD105, CD90, CD44 and negative for CD19; (B) Cumulative population doubling (CPD) of MSC expanded under normal and hypoxic conditions at p15; (E) Representative reverse-transcription polymerase chain reaction run on 2% of agarose gel showing expression of aP2 and adiponectin by adipocytes and RUNX2 and osteopontin by osteocytes cells, induced from human bone-marrow mesenchymal stem cells. (F) Senescence-associated βGal activity in MSC of normal vs Hypoxic at p15. Figure 4: Reproducibility of NGS and comparison of miRNA expression between NGS and qPCR analysis.

Click here for additional data file.

We would like to thank CryoCord Sdn. Bhd. for providing the bone marrow-derived mesenchymal stem cells.

Additional Information and Declarations

Competing Interests

Author Contributions

Data Deposition

Soon Keng Cheong is currently an employee of Cryocord Sdn Bhd; however, his position in CryoCord does not influence the design and output of the study. The rest of the authors declare that they have no competing interests.

Norlaily Mohd Ali conceived and designed the experiments, performed the experiments, analyzed the data, wrote the paper, prepared figures and/or tables, reviewed drafts of the paper.

Lily Boo performed the experiments, analyzed the data, reviewed drafts of the paper.

Swee Keong Yeap conceived and designed the experiments, analyzed the data, contributed reagents/materials/analysis tools, reviewed drafts of the paper.

Huynh Ky performed the experiments.

Dilan A Satharasinghe analyzed the data.

Woan Charn Liew performed the experiments.

Han Kiat Ong analyzed the data, contributed reagents/materials/analysis tools, reviewed drafts of the paper.

Soon Keng Cheong contributed reagents/materials/analysis tools.

Tunku Kamarul contributed reagents/materials/analysis tools.

The following information was supplied regarding data availability:

GEO accession number: GSE67630

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
