# Peer review of "Probable impact of age and hypoxia on proliferation and microRNA expression profile of bone marrow-derived human mesenchymal stem cells"

_PeerJ, doi:10.7717/peerj.1536_

## Round 0.1 · original submission · Major Revisions

The submission requires a major revision and I hope the comments will be taken care of as these comments will improve the manuscript in general.

Reviewer 1 ·

Basic reporting

No Comments

Experimental design

Methods:
The authors should provide a more detailled description of the differentiation assays for osteogenic and adipogenic differentiation, i.e. the composition of differentiation media.

The authors should state more clearly how the qPCR data was normalized. In the methods section, the geNorm algorithm is mentioned; in Table 1, sequences of miR-19a-3p and miR-200a-3p are given as reference genes. I assume that geNorm was used to identify stable reference gene candidates from NGS data. In the respective Figure (#3) it is not clear, whether data normalized to miR-19b-3p or miR-200a-3p is shown. The authors should provide arguments why standard reference genes such as U6 snRNA or 5S rRNA have not been used in parallel. Especially, since miR-19b-3p was found to be regulated, which is co-transcribed with miR-19a-3p as part of the miR-17-92 cluster. Also, it is known that miR-17-92 is downregulated during cellular senescence (Hackl et al Aging Cell, 2010)! The authors should further try to correlate NGS LogFCs to qPCR LogFCs and provide correlation coefficients.

Validity of the findings

The authors state in line 185ff that calcium deposition and lipid droplet deposition was reduced in aged hypoxic MSCs compared to young. This is also discussed in line 273ff. The evidence for altered differentiation capacity should be increased, for example by quantifying Alizarin Red and Oil Red or by providing (supplemental) information about biomarker expression such as Runx2, Osteocalcin, or Alkaline Phosphatase.

It was observed that hypoxia enhances proliferation of young as well as old primary BMSCs. However, the effect of hypoxia on miRNA transcription was observed to be quite different for young and old BMSCs. It would generally be interesting to provide data from an exploratory analysis of miRNA transcription in young/aged and hypoxic/normoxic samples in the form of a heatmap or PCA. Next it would be of interest to highlight the overlaps in regulated miRNAs, i.e. in the form of a Venn diagram. This overlap might contributed to the similar phenotype (enhanced proliferation) under hypoxic conditions.

Additional comments

The manuscript is well written and addresses an interesting question. The experiments are for the most part described well, with few exceptions (qPCR and differentiation assays), and the data are presented in a clear fashion.
If the major revisions regarding NGS and qPCR data analysis, as well as BMSC differentiation can be addressed by the authors, I recommend publication in PeerJ.

Reviewer 2 ·

Basic reporting

The study by Ali et al. identifies the impact of hypoxia and aging on miRNA profile of bone marrow-derived human MSCs (BM-hMSC). The study is important since the effect of hypoxia on BM-hMSC, derived from aged donors, is limited. The authors identifies that distinct population of miRNAs are expressed in hMSCs from young and aged people and associated miRNA profile with therapeutic potential of hMSCs. The study is generally well-written and precise but a few issues need to be fixed-.
1. The experiment mentioned in lines 185-189 of the manuscript as “data not shown” is crucial related to this study. Authors should present this result
2. The sentence between lines 51-54 must be rephrased
3. Line 121 should read as “The multipotency of MSCs....” and not “The multipotential of MSCs...”
4. Line 193 should be corrected to “...hypoxia-treated MSC”
Issues mentioned above and under the sections of "Experimental Design" as well as "Validity of the Findings" should be taken care of by the authors

Experimental design

1. It is well known that genetic variation and sex affect the prevalence and expression of miRNAs. There is no clue whether donors are of varied ethnicity/race or whether MSCs used in the study are sex-matched or not. Without this knowledge, logical interpretation does not seem possible. A three way analysis considering sex, race and age is recommended
2. N=3 per age group seems to be an insufficient sample size

Validity of the findings

1. The formula used to calculate population doubling is incorrect. The correct formula is PD=log(n2/n1)/log2, where n1 is the number of seeded cells and n2 is the number of cells harvested (EMBO Rep. 2009, 10, 71–78; Journal of Cell Science 2008, 121, 2235-2245). So, the data shown in Fig 1B seems problematic
2. Conclusions drawn on the impact of hypoxia and age on miRNA profile of hMSCs is overstated as those are prediction-based
3. Validity of findings are limited since the sample size is insufficient for this kind of studies

Additional comments

The miRNAs up or down-regulated in young and aged hypoxic and normoxic populations are known to have various functions. Unless experimental evidence-based linking of miRNAs with pathways is done, the study of “impact of age and hypoxia” remains at the level of mere predictions. This study truly “indicates” the “possible link” of hypoxia and aging on miRNA expression profile (quoted words are written in line 269 of the manuscript). Which is only “highly probable” (mentioned in line 285-286) cannot be reported as “impact”. Rather, the title should read as “Probable impact of age and hypoxia....”

---

## Round 0.2 · accepted · Accept

Dear Authors,

Your revised manuscript has been reviewed by two independent reviewers and both expressed satisfaction on the revised version. Based on their comments, I am happy to inform you that your revised manuscript has been now considered for the publication. However, you need to re-check your manuscript for possible grammar and spelling errors. You may get help from a native English speaker as well in this regard.

Reviewer 1 ·

Basic reporting

No comments

Experimental design

No comments

Validity of the findings

No comments

Additional comments

Thank you for addressing my comments! The manuscript is ready for publication.

Reviewer 2 ·

Basic reporting

The revised manuscript has no pending issues

Experimental design

The authors addressed raised issues properly

Validity of the findings

Validation has been done satisfactorily

Additional comments

Minor issues of spacing/formatting are noted (where new texts have been introduced). The authors should take care of this.